# Exploratory Role of Flavonoids on Metabolic Dysfunction-Associated Steatotic Liver Disease (MASLD) in a South Italian Cohort

**DOI:** 10.3390/antiox13111286

**Published:** 2024-10-24

**Authors:** Caterina Bonfiglio, Rossella Tatoli, Rossella Donghia, Davide Guido, Gianluigi Giannelli

**Affiliations:** 1Unit of Data Science, National Institute of Gastroenterology—IRCCS “Saverio de Bellis”, Castellana Grotte, 70013 Bari, Italy; rossella.donghia@irccsdebellis.it (R.D.); davide.guido@irccsdebellis.it (D.G.); 2Scientific Direction, National Institute of Gastroenterology—IRCCS “Saverio de Bellis”, Castellana Grotte, 70013 Bari, Italy; gianluigi.giannelli@irccsdebellis.it

**Keywords:** MASLD, flavonoids, Mediterranean diet

## Abstract

Background: Metabolic dysfunction-associated steatotic liver disease (MASLD) is the most recent definition for steatotic liver disease associated with metabolic syndrome. The results of recent metabolic and observational studies suggest a potential beneficial effect of food-derived flavonoids in some chronic diseases, including MASLD. The study aims to evaluate the protective role of diet flavonoids in subjects with and without MASLD belonging to a cohort living in the South of Italy. Methods: The study cohort comprised 1297 participants assessed in the NUTRIHEP cohort (2015–2018), divided into two groups, based on presence or absence of MASLD. Results: The results indicated statistically significant flavonoid consumption, showing a protective role against MASLD, at an optimal concentration of 165 mg/day, with an OR value of 0.63, (*p* = 0.001, 95% C.I.: 0.47; 0.83 t). The OR remained almost unchanged when the intake increased from 165 mg per day to 185 mg per day. Conclusions: In conclusion, our study results show a protective role of flavonoids against MASLD. Consuming only 165 mg of flavonoids daily can activate this protective function, reducing the risk of MASLD.

## 1. Introduction

Metabolic dysfunction-associated steatotic liver disease (MASLD) is the most recent definition for steatotic liver disease associated with metabolic syndrome [1].

In 1980, Ludwing et al. first reported the existence of a liver condition similar to alcoholic hepatitis diagnosed by ultrasound in moderately obese subjects who did not consume significant amounts of alcohol [2]. In 2007, Farrel et al. proposed the working definition of NAFLD for this condition [3]. The close association between NAFLD and metabolic syndrome soon led to the need for a change in the nomenclature [4]. MASLD was introduced in June 2023 with a multi-society Delphi consensus statement on a new fatty liver disease nomenclature that led to the final withdrawal of the term NAFLD [5]. Recent studies have suggested an association between NAFLD and metabolic syndrome [6].

MASLD is characterized by fat accumulation in the liver detected by imaging or biopsy and is observed in individuals with little or no alcohol consumption, generally affected by obesity, type 2 diabetes mellitus (T2DM), dyslipidemia, and/or hypertension [7]. Currently, the pathogenesis of MASLD is not well defined. Probably, a role is played by insulin resistance and interactions of genetic and environmental factors, as well as metabolic dysfunction [4]. Metabolic dysfunction-associated steatohepatitis (MASH), the more severe form of MASLD, is histologically defined by the presence of lobular inflammation and hepatocyte ballooning, and is associated with a greater risk of fibrosis progression [8].

MASLD is the most common cause of chronic liver disease and the leading cause of liver-related morbidity and mortality [1]. Cardiovascular diseases are the leading cause of death for patients with MASLD [9]. NAFLD affects about 30% of the world’s adult population, and the prevalence has risen from 22% to 37% in 20 years [10,11].

The prevalence of MSLD increases in parallel with the prevalence of obesity and related diseases, confirming the close connection between the two conditions [1]. The incidence of MASLD and MASH is strongly related to a sedentary lifestyle and excess dietary energy intake [12].

There is currently no Food and Drug Administration (FDA)-approved pharmacological therapy for MASLD and MASH, but a lifestyle intervention approach is suggested.

A randomized controlled trial showed the positive effect of a lifestyle modification program in patients with nonalcoholic fatty liver disease [13].

A healthy diet can significantly improve MASLD, based on the Mediterranean diet (MedDiet) model and associated with moderate physical activities [1,14].

The MedDiet owes its protective and preventive effects against various chronic diseases to the bioactive components in its foods [14]. These include vegetables and fruits, rich in fiber and polyphenols, fish, nuts, and extra virgin oil, rich in monounsaturated and polyunsaturated fatty acids [15]. Flavonoids are bioactive polyphenols ubiquitously present in plants [16,17].

The results of recent metabolic and observational studies suggest a potential beneficial effect of food-derived flavonoids in some chronic diseases, including NAFLD [16,17,18]. Many studies have been conducted in animal models to investigate the impact of flavonoids on fatty liver disease, but very few have been conducted in humans [18,19,20].

The scientific literature lacks studies evaluating the link between flavonoids consumption and MASLD.

The study aims to evaluate the protective role of diet flavonoids in subjects with and without MASLD belonging to a cohort living in the South of Italy.

## 2. Materials and Methods

### 2.1. Study Population

The NUTRIHEP study was a cohort created in 2005–2006 using the medical records of general practitioners in Putignano for persons aged 18 years and over. The study design involved selecting a sample of the general population aged 18 years and over through a systematic random sampling procedure from the lists of GP registers. Instead of using census data, we used the GP registers because no significant differences were found between the age and gender distribution of the general population of Putignano and that recorded in the GP registers. In Italy, since it is a legal requirement that everyone should have a general practitioner, the lists of the general population at GP surgeries correspond to the complete census list [21].

Trained physicians and/or nutritionists interviewed the participants at baseline in 2005–2006 to gather information on sociodemographic characteristics, health status, personal history, and lifestyle factors. This included a history of tobacco use, food intake, educational level [22], work profile [23], and marital status. Weight and height were measured with the participants wearing underclothing and no shoes. Weights were recorded to the nearest 0.1 kg using an electronic balance (SECA©), while height was recorded to the nearest 1 cm using a wall-mounted stadiometer (SECA©). Blood pressure (BP) was measured following international guidelines [24], and the average of three measurements was calculated.

The European Prospective Investigation into Cancer and Nutrition (EPIC) food frequency questionnaire (FFQ) was used to document the usual food intake of participants at baseline [25,26]. Nutritionists conducted an in-person structured interview asking participants to report on their frequency of usual intake of 260 food items over the past year; they reported intakes per day, per week, or per year. They were also asked to estimate their portion sizes from photographs; questions were referred to the usual intake in the last year.

From 2015 to 2018, all subjects participating in Nutrihep were recalled for the first follow-up of this cohort study. A total of 1426 subjects responded, and the respondents were subjected to the same protocol as the first enrolment.

All participants signed informed consent acknowledgements after receiving full information about their medical data to be studied. In this paper, we consider the data collected during the follow-up.

Eligible subjects were found to number 1297 (90.9%) (MASLD No, 668 subjects and MASLD Yes, 629 subjects) out of 1426 total. Eighty participants who had not completed the dietary questionnaire and 49 with hepatic steatosis and Hcv+ or Alcoholic Fatty Liver Disease were excluded (Figure 1).

The study was approved by the Ethical Committee of the Minister of Health (DDG-CE-792/2014, on 14 February 2014).

### 2.2. Outcome Assessment

As published in previous studies [1,27], the definition of MASLD was based on the presence of hepatic steatosis plus at least 1 of the following 5 conditions: (1) BMI > 25 kg/m^2^ or waist circumference > 94 cm in men and >80 in women; (2) fasting serum glucose ≥ 100 mg/dL (≥5.6 mmol/L), 2 h post-load glucose level ≥ 140 mg/dL (≥7.8 mmol/L), HbA1c ≥ 5.7%, or in specific drug treatment; (3) blood pressure ≥ 130/85 mmHg or in specific drug treatment; (4) plasma triglycerides ≥ 150 mg/dL (≥1.70 mmol/L) or in specific drug treatment; and (5) plasma HDL Cholesterol < 40 mg/dL (<1.0 mmol/L) for men and <50 mg/dL (<1.3 mmol/L) for women or in specific drug treatment.

Appendix A shows the distribution between MASLD No and MASLD Yes groups according to positive diagnostic criteria.

Furthermore, the definition of MASLD continued to limit alcohol intake (as previously for NAFLD) in the context of steatosis to an average daily intake of 20–50 g for women and 30–60 g for men [5].

Finally, other forms of liver disease coexisting with MASLD, such as HCV+, were ruled out to avoid altering the natural history of the disease [7].

Hepatic steatosis for individuals in the Nutrihep study was assessed.

### 2.3. Assessment of Flavonoid Intake

Flavonoid intake was calculated based on the daily consumption of the following healthy foods, typical of the Mediterranean diet: apples, grapes, oranges, orange juice, pears, peaches, strawberries, walnuts, broccoli, onion, cooked onion, aubergines, courgettes, spinach, celery, and tomato. The amount of flavonoids (mg/day) was obtained from the database Phenol-Explorer: an online comprehensive database on polyphenol contents in foods [28,29]. Appendix A displays the average amount of flavonoids (mg/day) and standard deviation (SD) in the health foods from the Nutrihep cohort.

### 2.4. Statistical Analysis

The individual characteristics are reported as means and standard deviations (M ± SD) or medians and interquartile ranges for continuous variables and as frequencies and percentages (%) for categorical variables.

We fitted a logistic regression model with MASLD as the outcome variable and flavonoid intake (both continuous and categorical) as predictors.

Flavonoid intake was categorized according to 10 mg/day intervals from 165 (lower) to 235 (upper) mg/day. Initially, confounding variables were selected from the existing literature. Then, the procedure of minimum absolute reduction and selection (LASSO) was adopted to reduce the number of candidate predictors and select those most useful for model construction (Appendix A) [30]. In selecting variables to be added to the model as confounders, those already included in the definition of MASLD, such as BMI, waist, HDL, triglycerides, glucose, HbA1c, and blood pressure, were not considered. The models were adjusted for the following variables: gender, age (<65 vs. ≥65 years), daily calories, weight (kg), γGT, ALT, HOMA (<2.5 vs. ≥2.5), job, and marital status. Estimated coefficients were transformed into odds ratios (OR).

In addition, the variance inflation factor (VIF) was also evaluated to check multicollinearity, and confounders with VIF > 5 were discarded (Appendix A) [31].

A forest plot was drawn to compare the OR values obtained by the model fitting, considering flavonoid consumption as a categorical variable.

In addition, dose–response modelling for dichotomous outcomes was fitted to determine the shape and magnitude of the relationship between exposure and MASLD [32] firstly, by considering the flavonoids intake categorized in deciles and secondly, by using a smoothing cubic spline function, with 9 knots on the percentiles 0% (or 1%), 12.5%, 25%, 37.5%, 50%, 62.5%, 75%, 87.5%, and 100% (or 99%) [33,34]. The reference value of the flavonoid intake regressor was the median. In this case, the odds ratios (ORs) of the dose–response relationships (with 95% confidence bounds, i.e., confidence intervals) were plotted in continuous shape around flavonoid intake (mg/day), and the evaluation in relation to OR = 1 was considered to denote significance or not. The adjustment covariates were selected by LASSO regression and fixed to the median values, mode, and reference category in relation to their continuous, categorical and dichotomous nature [35]. The two-tailed probability level was set at 0.05 to test the null hypothesis of non-association.

The analyses were conducted with StataCorp 2023 Stata Statistical Software: Release 18 (College Station, TX, USA: StataCorp LLC), while the forest plots [36] and dose–response modelling were created using RStudio and its packages forest plot, rms [35].

## 3. Results

Table 1 shows the main characteristics of the 1297 participants, classified according to MASLD (absence or presence); 48.50% of the sample had MASLD, most of whom (54.6%) were male.

Appendix A presents the main characteristics of the 1297 participants, categorized based on the median value of flavonoid intake (<185 vs. ≥185 mg/day).

The logistic regression models are reported in Table 2.

We found a statistically significant protective effect up to a maximum daily consumption of 225 mg, with an odds ratio (OR) of 0.74 (at 95% C.I. 0.56; 0.99).

The OR remained almost unchanged when the intake increased from 165 mg per day to 185 mg per day [OR: 0.63 (95% C.I. 0.47; 0.83) and 0.64 (95% C.I. 0.48; 0.85), respectively]. An increase by 20 mg daily did not change the protective effect against the risk of developing MASLD.

Appendix A shows the results of the regression model with the exposure variable divided into deciles and adjusted for the same variables as the models shown in Table 2, from which the statistically significant no longer protective effect is shown from decile 6 onward.

The distribution of deciles by the presence and absence of MASLD is shown in Appendix A.

In Figure 2, the protective role of flavonoids has been shown across various consumption categories.

Figure 2 was useful to represent, in a graphical manner, the estimation as odds ratios (ORs). It provides a visual way to compare the size and direction of the effects. On the right side of the plot, the numerical values of the ORs are present along with their confidence intervals at 95%, corresponding to each flavonoids category. Furthermore, the absolute frequency of subjects divided of MASLD categories was present. In this case, all OR values were <1.0, indicating the protective role of flavonoids intake, and the value of 1.00 was the null effect on MASLD outcome.

Figure 3 illustrates that the smoothing dose–response relationship of the daily flavonoid intake on MASLD is increasing up to about 240–250 mg, by indicating a protective role of flavonoid intake up 185 mg. However, the OR is significant (or quasi-) up to 80–90 mg. However, beyond 185 mg/day, the dose–response relationship is centered around OR = 1, indicating no effect.

Concerning that, the shaded area on the graph shows the confidence bands, representing the confidence intervals for each OR value. The OR is considered statistically significant for statistical significance if 95% C.I. does not include the value ‘1’. Notably, a large 95% confidence interval indicates a small sample size for the corresponding flavonoid intake values.

## 4. Discussion

This study, carried out in a population of 1297 Italian middle-aged participants from Putignano (Puglia, Italy), describes the effect of dietary flavonoid intake on the risk of MASLD.

Our results show that consumption of only 165 mg per day is already protective against MASLD. Since, until now, lifestyle intervention has been the cornerstone in the management of patients with MASLD, this finding may make an important contribution [1].

Given the recent emergence of MASLD, the scientific literature is lacking in studies evaluating the effect of bioactive components such as polyphenols on the risk of this liver condition. Studies evaluating the relationship with NAFLD are also very scarce and often report controversial results [37,38,39].

Given the recent new fatty liver disease nomenclature, from NAFLD to MASLD, we will refer to the studies in the scientific literature on NAFLD and flavonoids to discuss our findings.

Flavonoids are a large group of natural substances with variable phenolic structures [40]. They can be subdivided into flavonoids, flavonols, orange ketones, isoflavones, anthocyanins, chalcones, and dihydrogen derivatives [41]. These classes are amply present in fruits, vegetables, grains, bark, roots, stems, flowers, tea, and wine [40]. Flavonoids have various positive effects on human health, including antitumor, antioxidant, antibacterial, antiviral, anti-inflammatory, and analgesic effects [42,43]. These properties make flavonoids potentially beneficial substances for liver diseases as well [44].

The excessive intake of flavonoids may cause harmful effects that are not yet well known. At higher doses, flavonoids may act as mutagens, free radical-generating pro-oxidants, and inhibitors of key enzymes involved in several metabolic processes. Therefore, at high doses, the adverse effects of flavonoids may outweigh the beneficial ones. Although further studies are needed to precisely define a toxic dose, it is hypothesized that it is difficult to incur these harmful effects with the intake of dietary flavonoids alone [45,46].

Initially, flavonoids’ health benefits were attributed to their potent antioxidant capacity. However, other studies also suggest the importance of their anti-inflammatory and metabolic effects [18]. Flavonoids positively intervene in various forms of liver steatosis, such as by regulating lipid metabolism, insulin resistance, inflammation, and oxidative stress [45].

Wang et al. observed a reduction in hepatic fat content in mice treated with a high-fat diet after flavonoids from Broussonetia papyrifera treatment [47].

This flavonoid treatment also inhibited the production of ROS, reduced the content of myeloperoxidase, and improved the activity of superoxide dismutase (SOD). These results demonstrate the ability of flavonoids to reduce fat accumulation and oxidative stress.

Another study evaluated the effects of licorice chalcone [48]. This inhibits adipogenesis and increases lipid decomposition and fatty acid β-oxidation in fatty-liver mice by promoting the Sirtuin1/AMP-activated protein kinase pathway. Zhu et al. identified the same effect on lipid metabolism mediated by luteolin, lycopene, and their combination. They indirectly activate the SIRT1/AMPK pathway, thus inhibiting lipogenesis and increasing β-oxidation [49].

Yin et al. studied the effects of Cyanidin-3-O-glucoside, the most abundant anthocyanin in the flavonoid family, in fatty liver disease. They found that this kind of flavonoid is able to eliminate damaged mitochondria to maintain mitochondrial homeostasis and alleviate oxidative stress [50].

Positive effects on improving oxidative stress at the liver level are induced by quercetin, which has been demonstrated to restore the levels of superoxide dismutase, catalase, and glutathione in the liver of NAFLD mice [51]. Flavonoids may inhibit oxidative stress by regulating malondialdehyde (MDA), superoxide dismutase (SOD), and catalase (CAT) [52].

In liver steatosis, oxidative stress promotes inflammatory responses promoting liver injury. When the level of oxidative stress increases, it can promote the expression of IL-6, IL-1β, and TNF-α [53].

Wang et al. found that the levels of IL-1β, IL-6, and TNF-α, higher in the liver tissue of rats in the NAFLD model group, could be reduced by the total flavonoids of *Scutellaria baicalensis* [52]. Subsequent studies have found the same results [54]. These results support the idea that the anti-inflammatory effect of flavonoids occurs mainly through the inhibition of the NF-κβ pathway [55]. Luteolin has also been shown to reduce a variety of inflammatory factors in rats with liver steatosis, proving to have not only an antioxidant effect but also a good anti-inflammatory effect [56]. Thus, the main pathways through which flavonoids exert their protective effect toward liver steatosis are anti-inflammatory and antioxidant [57].

They can also decrease oxidative damage through a free radical scavenging activity because they have hydroxyl groups [58].

The main food source of flavonoids for the human body is fruits and vegetables [40]. Our results are in line with other studies that have reported a link between vegetable and fruits intake and a lower risk of liver steatosis [44,59].

They are also in agreement with the results that suggested that the Mediterranean diet’s beneficial effect is due to its high content of bioactive phytochemicals [60]. Fruits and vegetables are at the base of the Mediterranean diet food pyramid, which includes their consumption at every main meal [61]. Our study population is strongly tied to the Mediterranean food culture and habitually eats foods typical of this dietary pattern.

Consuming one apple per day is sufficient to achieve the flavonoid intake we have identified as protective toward MASLD [27]. The apple is now readily available all year round and it is affordable, allowing the entire population to consume it regularly.

### Strengths and Limitations

The present study evaluated the effect of dietary flavonoid intake and risk of MASLD in a middle-aged cohort from Putignano.

The strengths of the present study include the generalizability of the results to southern Mediterranean populations. The main strength of our study is that no other study in the scientific literature has analyzed this aspect of MASLD, nor has it done so in a similar population. However, some limitations must be considered. One of the main limitations is the absence of data on individual flavonoid classes. This did not allow us to investigate whether there is a class of flavonoids that has a greater protective effect toward MASLD than others. Another potential limitation is related to the self-reporting of the diet, although to remedy canvas, each FFQ was checked by our dietitian upon delivery of the questionnaire. We were also unable to examine the effect of parsley consumption on the risk of MASLD. Parsley is rich in flavonoids [62] and frequently used as an aromatic seasoning herb in the South of Italy.

## 5. Conclusions

In conclusion, our study results show a protective role of flavonoids against MASLD. Consuming only 165 mg of flavonoids daily can activate this protective function, reducing the risk of MASLD.

However, further studies are needed to support the validity of our results also in different populations, identify the classes of flavonoids with potentially greater protective roles, and define the proportions of single classes to be eaten that may enhance the protective effect.

## Figures and Tables

**Figure 1 antioxidants-13-01286-f001:**
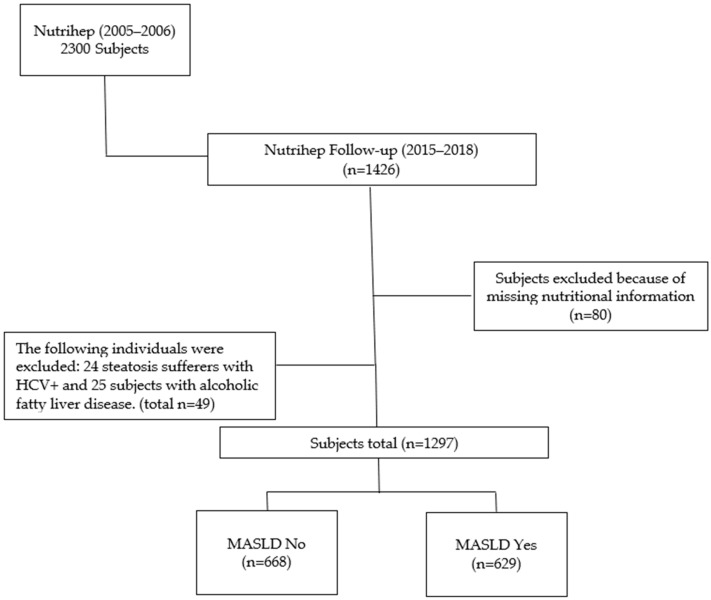
Flow chart.

**Figure 2 antioxidants-13-01286-f002:**
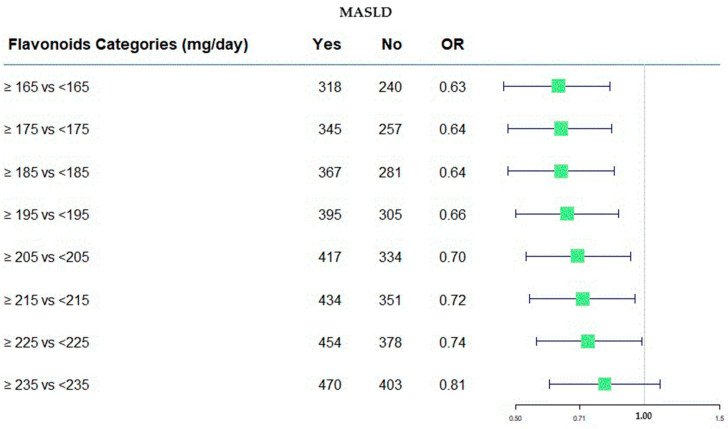
Forest plot of OR values in the MASLD by flavonoid categories (mg/day).

**Figure 3 antioxidants-13-01286-f003:**
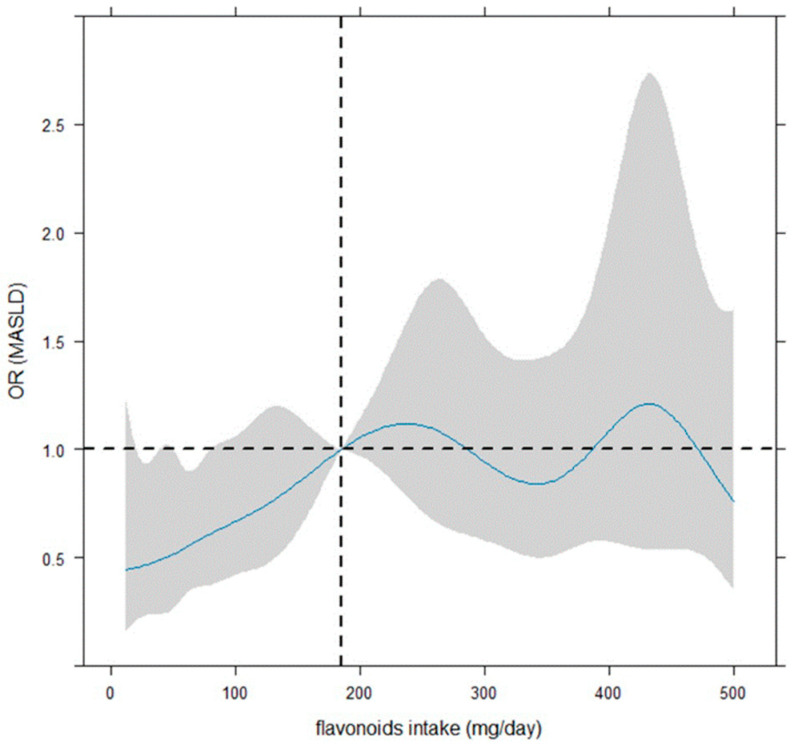
Dose–response curves for flavonoid intake-MASLD by restricted cubic spline. The reference value of the flavonoid intake regressor was the median (185 mg/day, vertical dotted line). The odds ratios (ORs) of the dose–response relationship (with 95% confidence bounds shown as shaded zones) were plotted in continuous shape around flavonoid intake (mg/day), and the evaluation in relation to OR = 1 (i.e., horizontal dotted line) was considered to denote significance. The logistic model was adjusted for job, daily calories, weight (kg), gamma glutamyl transferase, alanine amino transferase, gender (female vs. male), age (<65 vs. ≥65 years), marital status, and HOMA (<2.5 vs. ≥2.5).

**Table 1 antioxidants-13-01286-t001:** Characteristics of participants by MASLD. Nutrihep study. Putignano (BA). Italy. 2015–2018.

Parameters ^a^		MASLD	
	Whole Sample ^b^	No	Yes
N (%)	1297	668 (51.50)	629 (48.50)
Age (years)	54.33 (14.34)	49.24 (13.80)	59.74 (12.86)
Age categories (years) (%)			
<65	955 (73.6)	565 (59.2)	390 (40.8%)
≥65	342 (26.4)	103 (30.1)	239 (69.9%)
Gender (%)			
Female	744 (57.4)	417 (56.0)	327 (44.0%)
Male	553 (42.6)	251 (45.4)	302 (54.6%)
Flavonoids (mg/day)	203.89 (126.36)	192.30 (123.03)	216.19 (128.75)
rMED (median (IQR))	8.00 (6.00; 10.00)	8.00 (6.00; 10.00)	8.00 (6.00; 10.00)
BMI (kg/m^2^)	27.58 (5.05)	25.04 (3.59)	30.28 (4.97)
Weight (kg)	72.93 (14.87)	66.66 (12.02)	79.58 (14.73)
Waist (cm)	90.45 (13.46)	83.04 (10.38)	98.32 (11.79)
SBP (mmHg)	120.93 (15.81)	115.64 (15.35)	126.52 (14.30)
DBP (mmHg)	77.68 (8.00)	75.69 (7.88)	79.78 (7.58)
HbA1c (mmol/mol)	38.07 (6.87)	36.59 (5.05)	39.64 (8.09)
HOMA	1.89 (1.88)	1.33 (0.90)	2.43 (2.38)
ALT (U/L)	22.20 (16.21)	19.70 (8.27)	24.86 (21.37)
γGT (U/L)	17.58 (13.46)	14.80 (7.67)	20.54 (17.16)
AST (U/L)	21.74 (10.87)	20.70 (5.94)	22.85 (14.29)
TG (mg/dL)	98.41 (69.23)	80.73 (58.55)	117.22 (74.60)
C-reactive protein (mg/dL)	0.26 (0.55)	0.21 (0.52)	0.31 (0.58)
TC (mg/dL)	191.35 (35.36)	188.90 (33.06)	193.96 (37.50)
HDL (mg/dL)	50.79 (12.59)	53.18 (12.80)	48.24 (11.85)
Glucose (mg/dL)	95.34 (17.34)	90.13 (10.54)	100.89 (21.06)
ALP (U/L)	52.98 (16.10)	50.10 (15.56)	56.04 (16.11)
Alcohol intake (g/day)	10.58 (12.72)	10.74 (13.41)	10.42 (11.96)
Kcal (day)	2056.26 (750.22)	2100.33 (724.88)	2009.46 (774.05)
Smoker (%)			
Never/Former	1137 (87.7)	587 (51.6)	550 (48.4)
Current	159 (12.3)	81 (50.9)	78 (49.1)
Hypertension (%)			
No	847 (68.8)	517 (61.0)	330 (39.0)
Yes	385 (31.2)	115 (29.9)	270 (70.1)
Dyslipidemia (%)			
No	1047 (85.1)	561 (53.6)	486 (46.4)
Yes	184 (14.9)	71 (38.6)	113 (61.4)
Diabetes (%)			
No	1148 (93.2)	620 (54.0)	528 (46.0)
Yes	84 (6.8)	12 (14.3)	72 (85.7)
Marital Status (%)			
Single	181 (14.0)	115 (63.5)	66 (36.5)
Married or living together	1034 (79.7)	519 (50.2)	515 (49.8)
Separated or divorced	28 (2.2)	20 (71.4)	8 (28.6)
Widow/er	54 (4.2)	14 (25.9)	40 (74.1)
Education (%)			
Primary school	282 (21.8)	71 (25.2)	211 (74.8)
Secondary school	383 (29.5)	171 (44.6)	212 (55.5)
High school	460 (35.5)	307 (66.7)	153 (33.3)
Graduate	172 (13.3)	119 (69.2)	53 (30.8)
Job (%)			
Managers and professionals	102 (7.9)	57 (55.9)	45 (44.1)
Craft, agricultural, and sales Workers	469 (36.2)	285 (60.8)	184 (39.2)
Elementary occupations	185 (14.1)	93 (50.3)	92 (49.7)
Housewife	141 (10.9)	74 (52.5)	67 (47.5)
Pensioner	325 (25.1)	110 (33.8)	215 (66.2)
Unemployed	75 (5.8)	49 (65.3)	26 (34.7)
Family income assessment (%)			
Insufficient	27 (2.1)	10 (37.0)	17 (63.0)
Just sufficient	167 (12.9)	81 (48.5)	86 (51.5)
Sufficient	1019 (78.6)	521 (51.1)	498 (48.9)
More than sufficient	64 (4.9)	44 (68.8)	20 (31.2)
Good	20 (1.5)	12 (60.0)	8 (40.0)

^a^ As means and standard deviations. MASLD: metabolic dysfunction-associated steatotic liver disease; rMED: relative Mediterranean diet; BMI: body mass index; SBP: systolic blood pressure; DBP: diastolic blood pressure; HbA1c: glycosylated hemoglobin; HOMA: homeostasis model assessment; ALT: alanine amino transferase; γGT: gamma glutamyl transferase; AST: aspartate amino transferase; TG: triglycerides; TC: total cholesterol; HDL: high-density lipoprotein cholesterol; ALP: alkaline phosphatase level. ^b^ Percentages calculated for the column. Otherwise, percentages are calculated for the row.

**Table 2 antioxidants-13-01286-t002:** Logistic regression analysis of MASLD on flavonoids intake as continuous and categorical variables inserted in the models.

Flavonoids (mg/day)	MASLD			
	OR	SE (OR)	*p-Value*	95% C.I.
Categories ^b^:				
≥165 vs. <165	0.63	0.09	0.001	0.47; 0.83
≥175 vs. ≤175	0.64	0.09	0.002	0.48; 0.84
≥185 vs. <185 ^a^	0.64	0.09	0.002	0.48; 0.85
≥195 vs. <195	0.66	0.09	0.004	0.50; 0.87
≥205 vs. <205	0.70	0.10	0.014	0.53; 0.93
≥215 vs. <215	0.72	0.10	0.023	0.54; 0.95
≥225 vs. <225	0.74	0.11	0.045	0.56; 0.99
≥235 vs. <235	0.81	0.12	0.170	0.60; 1.09
continuous	1.001	0.001	0.016	1.000; 1.003

^a^ Median value. ^b^ ≥referent categories. Models adjusted for job, daily calories, weight (kg), gamma glutamyl transferase, alanine aminotransferase, gender (female vs. male), age (<65 vs. ≥65 years), marital status, and HOMA (<2.5 vs. ≥2.5). SE: standard error; MASLD: metabolic dysfunction-associated steatotic liver disease; OR: odds ratio.

## Data Availability

The original contributions presented in this study are included in the article. Further inquiries can be directed to the corresponding author.

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
