# Peer review of "Exploratory Role of Flavonoids on Metabolic Dysfunction-Associated Steatotic Liver Disease (MASLD) in a South Italian Cohort"

_antioxidants, 2024, doi:10.3390/antiox13111286_

Round 1
Reviewer 1 Report
This manuscript attempts to investigate the potential protective effects of dietary flavonoid intake on MASLD. While the study addresses an important public health issue and utilizes a substantial cohort, several methodological and reporting issues significantly undermine the robustness and validity of the findings. The paper would benefit from major revisions, particularly in clarifying its methodology, addressing confounding factors, and improving the clarity of its statistical analysis.
1. The manuscript frequently refers to MASLD, but it should provide a clearer distinction between MASLD and the formerly used NAFLD terminology. The transition from NAFLD to MASLD in the literature should be discussed more thoroughly.
2. The study mentions flavonoid intake but does not categorize or differentiate between types of flavonoids, which could have varying biological effects. This lack of specificity diminishes the depth of the analysis.
3. The cohort is derived from a specific region in Italy, which could introduce regional dietary habits that are not generalizable. The manuscript does not sufficiently address how this geographic specificity might affect the external validity of the results.
4. The use of logistic regression models is appropriate, but the justification for selecting particular confounding variables (e.g., Job, Daily calories) is not clear. Additionally, the manuscript does not address potential multicollinearity sufficiently, despite mentioning the variance inflation factor (VIF).
5. The study controls for some confounders but omits others that are critical in studying liver disease, such as alcohol consumption patterns (beyond daily intake averages), genetic predispositions, and detailed physical activity levels.
6. The interpretation of the dose-response relationship is ambiguous. The manuscript claims a protective effect at certain levels of flavonoid intake but fails to explain why higher intakes do not correspond with stronger protective effects, which contradicts expectations.
7. The manuscript relies on a food frequency questionnaire (FFQ) to assess dietary intake. However, it does not discuss the validation of this FFQ in the context of flavonoid intake or its potential limitations, such as recall bias.
8. There are inconsistencies in the description of the methods, such as unclear criteria for participant inclusion and exclusion, and how potential confounders were handled during analysis.
9. The conclusion that 165 mg/day of flavonoids is universally protective is overstated, given the study's regional and methodological limitations. The manuscript should emphasize that further research is needed to confirm these findings in diverse populations.
10. The manuscript mentions that the NUTRIHEP cohort was created in 2005-2006, but it also refers to data collection from 2015-2018. The relationship between these timeframes is unclear (lines 88, 119).
11. The study design is referred to as cross-sectional (line 124), which is not fully appropriate if follow-up data were involved.
12. The manuscript claims to have fitted a logistic regression model with flavonoid intake as a predictor (lines 164-166), but the description lacks detail on whether interactions or confounders were appropriately handled.
13. The manuscript provides Odds Ratios (ORs) but does not consistently report the associated confidence intervals (CIs) in every case, particularly in the abstract and some results sections (lines 23-26, 223-226).
14. The dose-response relationship is described as "centered around OR=1, indicating no effect" (line 235). This suggests no relationship, yet the discussion implies protective effects.
15. Figure 2 (Forest plot) and Figure 3 (spline plot) are referenced without clear explanations of their significance or interpretation in the text (lines 232-246).
16. The limitations mentioned do not address potential biases related to dietary self-reporting, which is a critical aspect in studies of this nature (lines 337-348).
Author Response
Si prega di consultare l'allegato

Reviewer 2 Report
This article by Bonfiglio is the first human long-term study to investigate the role of flavonoids in MASLD. The study is well-conceived and organized. The number of responders after 10 years is remarkable.
One of the biggest concerns I have with this study is the defining of people who have or do not have MASLD. Parameters of MASLD are defined in Lines 130-137 along with the exclusion of excessive alcohol and viral infections. Given these parameters, how many of the individuals in the study met all the criteria including waist circumference, fasting serum glucose, glucose tolerance, HBA1c, blood pressure, triglyceride levels, or HDL? I am assuming with a population of 1297 that not every single person in the "No MASLD" group had passed with all 7 parameters and that most people would be heterogenous such as having normal fasting glucose along with high triglycerides. Please explain the demarcation of the Yes and No groups.
Figure 2 is out of alignment with the text. Table 2 legend is out of alignment. These are probably technical errors during uploading.
Author Response
Si prega di consultare l'allegato
